# Landslide Susceptibility Mapping Based on Weighted Gradient Boosting Decision Tree in Wanzhou Section of the Three Gorges Reservoir Area (China)

**Yingxu Song** [1,2,†] , **Ruiqing Niu** [1,*], **Shiluo Xu** [3,†], **Runqing Ye** [4], **Ling Peng** [5], **Tao Guo** [6] , **Shiyao Li** [1] **and Tao Chen** [1]

1   Institute of Geophysics and Geomatics, China University of Geosciences (Wuhan), Wuhan 430074, China; yxsong@cug.edu.cn (Y.S.); li_sy91@126.com (S.L.); taochen@cug.edu.cn (T.C.)
2   Central-south China Geoscience Innovation Center, China Geological Survey, Wuhan 430205, China
3   School of Information Engineering, Huzhou University, Huzhou 313000, China; xushiluo@163.com
4   Wuhan Geological Survey Center, China Geological Survey, Wuhan 430205, China; yerunqing2005@126.com
5   China Institute of Geo-Environment Monitoring, Beijing 100081, China; pengl@mail.cigem.gov.cn
6   Sichuan Zhitu Information Technology Co., Ltd., Chengdu 610000, China; guotao0628@outlook.com
*   Correspondence: rqniu@163.com; Tel.: +86-133-7788-7265
†   These authors contributed equally to this work.

**Abstract:** The main goal of this study is to produce a landslide susceptibility map in the Wanzhou section of the Three Gorges reservoir area (China) with a weighted gradient boosting decision tree (weighted GBDT) model. According to the current research on landslide susceptibility mapping (LSM), the GBDT method is rarely used in LSM. Furthermore, previous studies have rarely considered the imbalance of landslide samples and simply regarded the LSM problem as a binary classification problem. In this paper, we considered LSM as an imbalanced learning problem and obtained a better predictive model using the weighted GBDT method. The innovations of the article mainly include the following two points: introducing the GBDT model into the evaluation of landslide susceptibility; using the weighted GBDT method to deal with the problem of landslide sample imbalance. The logistic regression (LR) model and gradient boosting decision tree (GBDT) model were also used in the study to compare with the weighted GBDT model. Five kinds of data from different data source were used in the study: geology, topography, hydrology, land cover, and triggered factors (rainfall, earthquake, land use, etc.). Twenty nine environmental parameters and 233 landslides were used as input data. The receiver operating characteristic (ROC) curve, the area under the ROC curve (AUC) value, and the recall value were used to estimate the quality of the weighted GBDT model, the GBDT model, and the LR model. The results showed that the GBDT model and the weighted GBDT model had a higher AUC value (0.977, 0.976) than the LR model (0.845); the weighted GBDT model had a little higher AUC value (0.977) than the GBDT model (0.976); and the weighted GBDT model had a higher recall value (0.823) than the GBDT model (0.426) and the LR model (0.004). The weighted GBDT method could be considered to have the best performance considering the AUC value and the recall value in landslide susceptibility mapping dealing with imbalanced landslide data.

**Keywords:** landslide susceptibility mapping; Three Gorges area; weighted GBDT; logistic regression; imbalanced landslide data

## 1. Introduction

Landslide is one of the most common geological hazards in the world with a high frequency, a wide distribution range, and serious disaster consequences, causing many casualties every year [1,2]. Landslide susceptibility mapping (LSM) is one of the most important methods in landslide hazard management and predicting the occurrence of the landslide, for the landslide is a complexed non-linear system [3–5]. LSM aims to establish the relationship between landslide location and its related factors to identify the areas that are prone spatially [3]. Many methods have been proposed in LSM with the development of geographic information systems (GIS) and remote sensing (RS) in the past 10 years [3,6]. Generally speaking, these methods can be divided into knowledge-driven, data-driven, and a combination of both. Representative knowledge-driven methods include fuzzy logic [7–15], fuzzy comprehensive evaluation, the analytic hierarchy process [16–18], and the expert system method, while data-driven models mainly include the information value [19], logistic regression [20–27], artificial neural networks [28–30], support vector machines [31–34], and other machine learning methods. The study in [3] found that the logistic regression model seemed to be the most popular method in LSM, which was also used in this study.

With the development of ensemble learning, the bagging and boosting methods are increasingly used for classification and regression. The ensemble learning approach has also achieved excellent results in many machine learning competitions. There are also studies using ensemble learning methods for LSM [4,35]. However, the GBDT method is rarely used in LSM compared to AdaBoost, random forest (RF) being one of the ensemble learning methods. In addition, in a given study area, the number of landslide samples is often much smaller than the number of non-landslide samples. When the sample division of the landslide was carried out, few studies paid attention to the problem of imbalanced landslide samples. The usual practice has been randomly resampling non-landslide samples so that the landslide samples have the same order of magnitude as the non-landslide samples, which always leads to a waste of non-landslide data and low predictive ability. Therefore, the non-landslide sample data are often wasted, resulting in the poor prediction ability of the model.

The study area suffers from extreme rainfall events because of the local climate, and the Three Gorges area in China has been seriously affected by landslide disasters and experienced significant and widespread landslide events in recent years [34]. The work in [36] used the information value method to calculate the landslide susceptibility in Wanzhou District. A self-organizing map network method was used in [37] to calculate the LSM in Wanzhou District. A weighted information value method was used to calculate the landslide susceptibility in Wanzhou District in [38]. In this paper, we studied the performance of the GBDT method for LSM in the study area. At the same time, considering the imbalance of landslide samples, a cost matrix was applied to the GBDT model, that is different weights were given to landslide samples and non-landslide samples to separate landslide samples as much as possible. Finally, a weighted GBDT model with a higher recall value and AUC value was obtained.

## 2. Materials and Methods

### 2.1. Study Area

Wanzhou is a district of Chongqing (China), bordering the northwest of Sichuan Province and the southeast of Hubei Province. It is also one of the major port cities in the Yangtze River Basin and an important industrial, cultural, trade, and transportation center in Yudong area [39]. Wanzhou District lies between latitudes $30°23'50''$ N and $31°0'18''$ N and longitudes $107°52'22''$ E and $108°53'52''$ E. The study area (Figure 1) belongs to the bank section of Wanzhou District with an area of 552 km$^2$, is distributed along the Yangtze River, and contains 202 historical landslides.

From a geological point of view, the area is mainly composed of Triassic and Jurassic formations, with a formation time of 230–137 million years ago. There are also Paleozoic Permian strata in the local

area from 285–230 million years ago, as well as the Cenozoic Quaternary strata of 2.5 million years ago. Figure 2 shows the distribution of the strata in the study area, with a scale of 1:50,000 [39].

Wanzhou District belongs to the subtropical monsoon humid zone with a mild climate and abundant rainfall. The annual average precipitation is 1191.3 mm, and almost 70% of the annual precipitation falls from May–September [39].

The main geological disasters in Wanzhou District include collapse, landslides, and bank collapse [38]. According to the landslide classification of USGS (https://pubs.usgs.gov/fs/2004/3072/pdf/fs2004-3072.pdf), the landslides in the study area mainly belong to block slide and a small part of the landslides belong to earth flow. The composition of the landslides are mostly Jurassic red slabs, gravel, and soil.

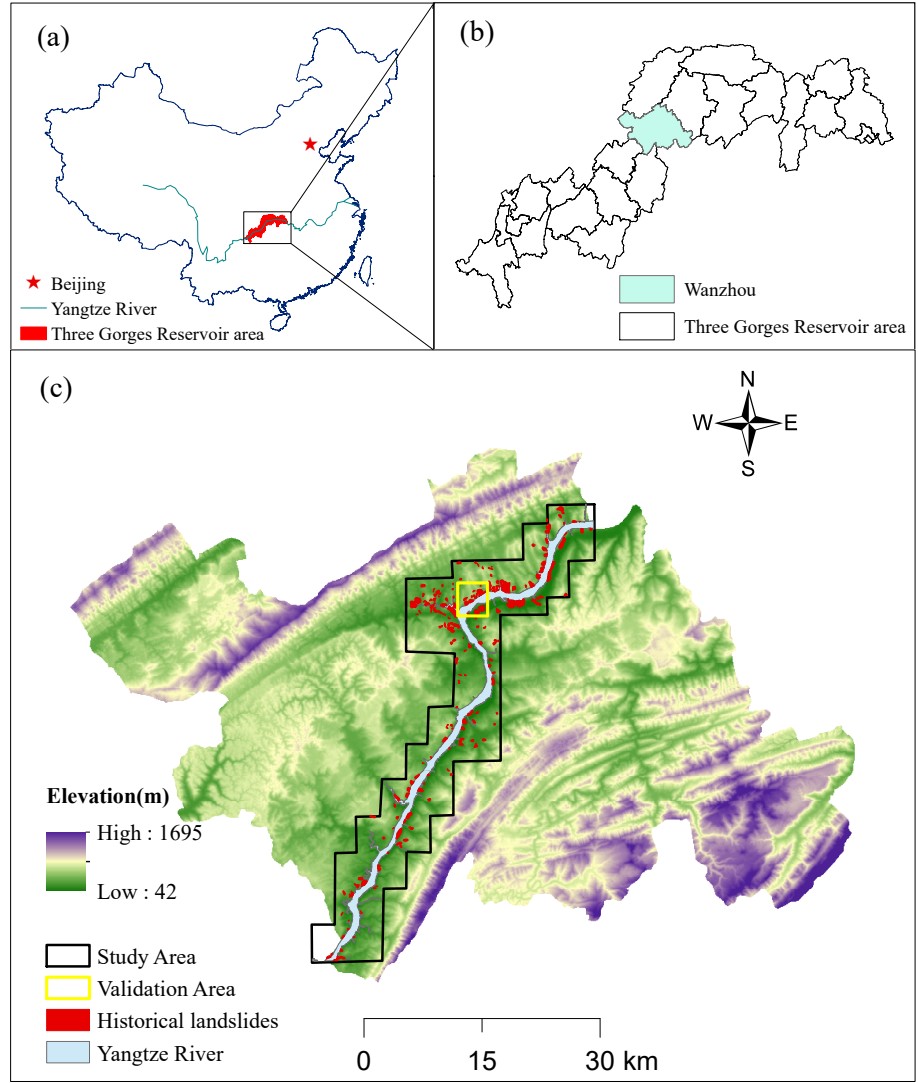

**Figure 1.** Location of the study area. (**a**) China. (**b**) Three Gorges Reservoir area. (**c**) Wanzhou District;the study area is in black outline. The yellow border indicates the verification area.

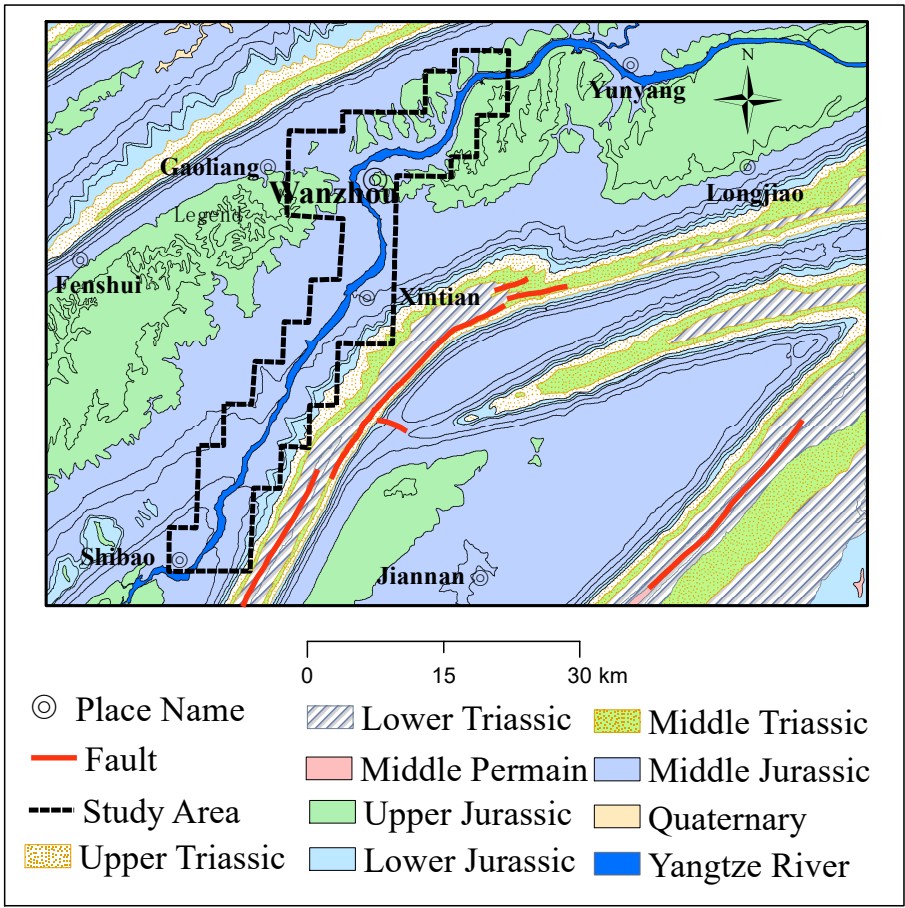

**Figure 2.** Stratigraphic map of the study area.

A digital elevation model (DEM) with 30 × 30 m resolution was used to produce a set of topographic factors, which is downloaded from http://www.gscloud.cn. A Landsat-8 satellite image acquired on 8 December 2013 was provided by the United States Geological Survey (USGS) (download from https://earthexplorer.usgs.gov/). Lithology data were collected from the local Land and Resources Bureau. Meteorological data were collected and compiled from the government of the Meteorological Bureau (downloaded from http://www.cma.gov.cn/). All of the mentioned data were processed and used to create the landslide-related factors with ArcGIS 10.2, QGIS 2.18.22, and SAGA 7.0.0. The logistic regression method and the GBDT method were supported by sklearn [40], which is an open source Python library.

*2.2. Landslide Susceptibility Mapping Process*

Our overall workflow can be seen in Figure 3. A total of 29 factors could be divided into 5 categories (Table 1).

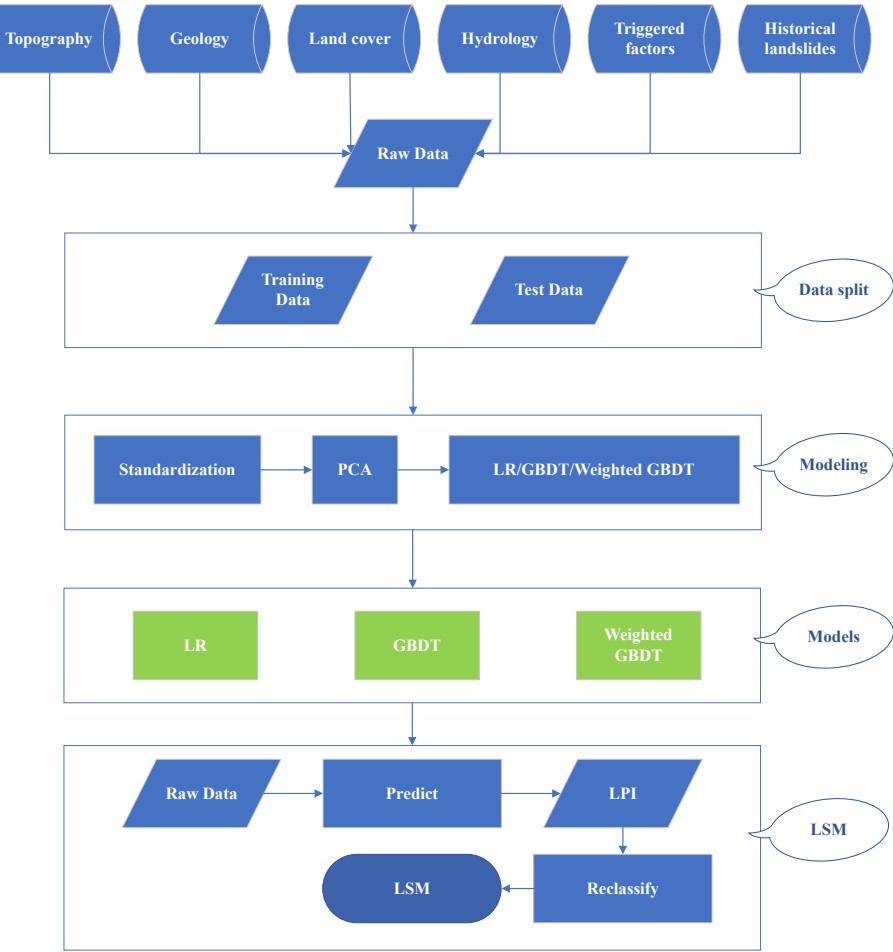

**Figure 3.** Overall workflow of this study.

In the evaluation of landslide susceptibility, many features do not mean good results [41]. Screening for landslide factors, removing multicollinearity between factors, and reducing the dimension of features helps to increase the robustness of the model.

A variety of methods have been used for data dimensionality reduction. Such as the certainty factor (CF) [42], principal components analysis (PCA), Pearson correlation (PC), and recursive feature elimination (RFE).

In this paper, we used the PCA method to reduce the dimensionality of the data. Eight factors with high explained variance were selected as input factors for landslide susceptibility evaluation. Figure 4 shows the distribution of eight landslide factors.

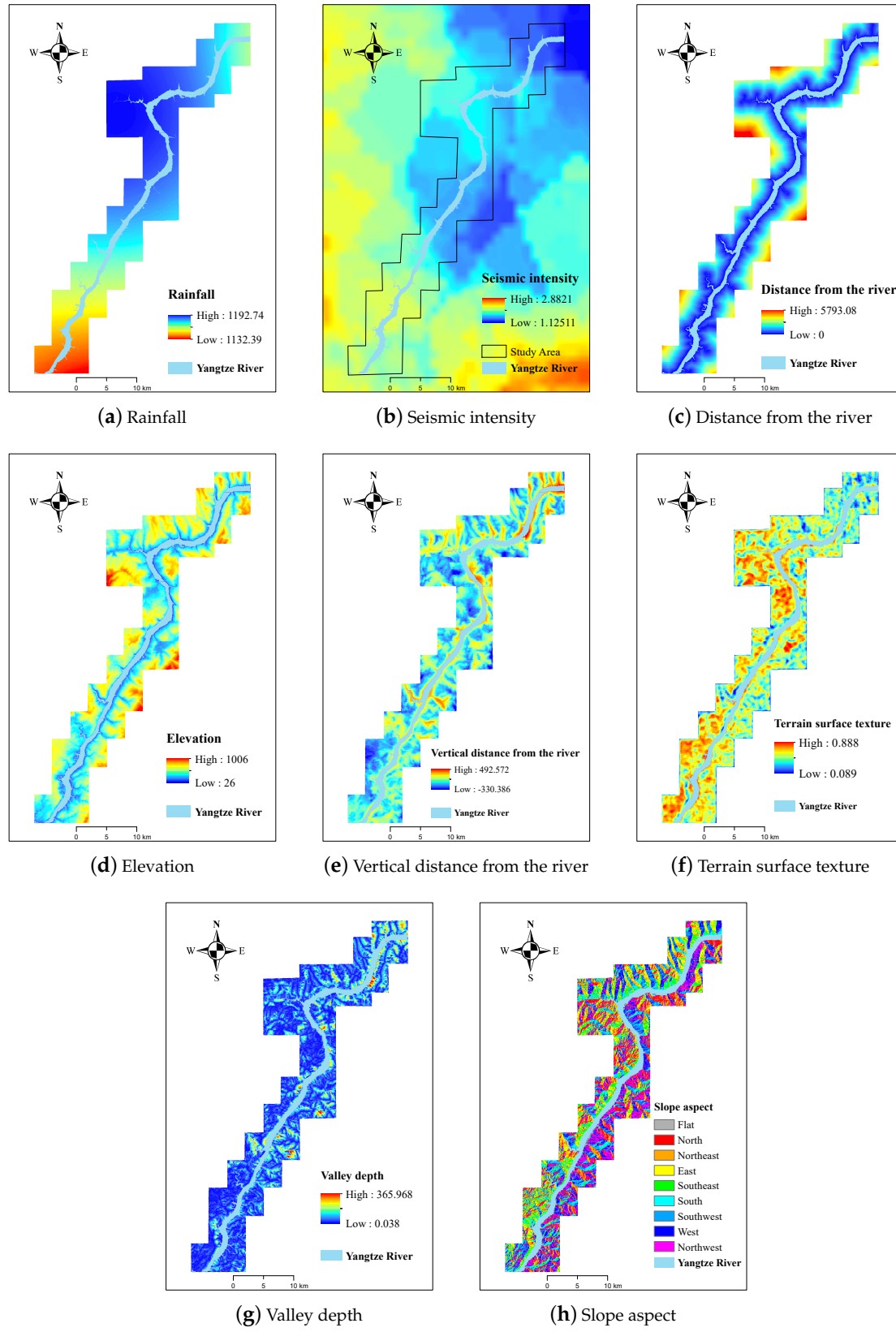

**Figure 4.** Landslide factors used in the study. (**a**) Rainfall (mm/year). (**b**) Seismic intensity. (**c**) Distance from the river (m). (**d**) Elevation (m). (**e**) Vertical distance from the river (m). (**f**) Terrain surface texture. (**g**) Valley depth (m). (**h**) Slope aspect.

**Table 1.** The names, descriptions, and classification of input variables.

| Variables | Name | Data Description | Class |
|---|---|---|---|
| Y | Landslide | Landslide occurred or not | Landslide |
| X1 | Elevation | Height above sea level | Topography |
| X2 | Slope angle | Extracted from DEM | Topography |
| X3 | Slope aspect | Extracted from DEM | Topography |
| X4 | Slope height | Extracted from DEM | Topography |
| X5 | Slope form | Extracted from DEM | Topography |
| X6 | TST | Terrain surface texture | Topography |
| X7 | TRI | Terrain roughness index | Topography |
| X8 | TPI | Terrain position index | Topography |
| X9 | TSC | Terrain surface curvature | Topography |
| X10 | TCI | Terrain convergence index | Topography |
| X11 | RSP | Relative slope position | Topography |
| X12 | Plain curvature | Extracted from DEM | Topography |
| X13 | Profile curvature | Extracted from DEM | Topography |
| X14 | Topographic curvature | Extracted from DEM | Topography |
| X15 | Geologic time | Different times of rocks and stratum | Geology |
| X16 | Slope structure | Structure of the slope | Geology |
| X17 | River | Distance from the river | Hydrology |
| X18 | Drainage area | Area of the drainage | Hydrology |
| X19 | Flow path length | Length of flow path | Hydrology |
| X20 | THI | Terrain humidity index | Hydrology |
| X21 | Valley depth | Depth of valley | Hydrology |
| X22 | Watershed slope | Slope of the watershed | Hydrology |
| X23 | RSI | River strength index | Hydrology |
| X24 | VDR | Vertical distance from the river | Hydrology |
| X25 | NDVI | Normalized Difference Vegetation Index | Land cover |
| X26 | NDWI | Normalized Difference Water Index | Land cover |
| X27 | Rainfall | Average annual rainfall | Triggered factors |
| X28 | Seismic intensity | Regional investigation intensity | Triggered factors |
| X29 | Land use | Land use classes | Triggered factors |

The main idea of landslide susceptibility mapping is as follows: 8 evaluation factors are used as condition attributes; the landslide samples are used as decision attributes; the condition attributes and decision attributes are formed into an initial decision table; then, the attribute is used as the input feature set of the model to train three models; finally, the landslide susceptibility mapping is realized. The detailed steps are as follows:

1.  Divide the model unit: The grid unit was used as the model unit in this study. The spatial resolution of the remote sensing image data and the DEM data was 30 m, so all of the evaluation factors were resampled to 30 m. The study area was divided into 582,486 cells, 553,172 of which were non-landslides, and the remaining 29,313 cells were landslides. The number of non-landslide cells was approximately 19-times the number of landslide cells.
2.  Construct an initial decision table: The condition attribute corresponding to 8 evaluation factors and the decision attribute corresponding to the landslide (1 represents landslide, 0 represents non-landslide) formed a two-dimensional table; each line describes an object, and each column corresponds to an attribute of the object. That is, the two-dimensional table contains 582,486 rows and 30 columns.
3.  The two-dimensional table was randomly divided into two parts: training data (70 percent) and test data (30 percent). The training data were used to build the model and the test data were used to make predictions.
4.  Evaluation of landslide susceptibility: All the model units in the study area were calculated by using the above three models, and the probability values of each model unit belonging to each category were output to generate the landslide prediction index (LPI) maps.

5.  Reclassify the LPI maps: The LPI maps were classified into five categories according to the natural breakpoint method; they were very high, high, moderate, low, and very low, meaning the level of landslide susceptibility.
6.  Analysis of the results. The three models were comprehensively evaluated using the receiver operating characteristic (ROC) curve, the area under the ROC curve (AUC) value, and the recall value. The reason for not using the Accuracy value is also given in this article.

### 2.3. Logistic Regression

Logistic regression is always used for dichotomous dependent or predictor variables [43]. The variables could be either continuous or discrete compared with the usual linear regression model [44]. The Landslide occurrence ($Y$) and landslide factors ($X_1$, $X_2$, ..., $X_n$) with an LR model can be expressed using Equation (1).

$$Y = \ln(\frac{p_i}{1 - p_i}) = \beta_0 + \beta_1 * X_1 + \beta_2 * X_2 + ... + \beta_n * X_n \tag{1}$$

where $p_i$ is the probability of $Y$ occurring at location $i$, $p_i/(1 - p_i)$ is the "odds ratio" or likelihood ratio, and $\beta_i(i = 0, 1, 2, ..., n)$ is the regression coefficient of the LR model [19].

Then, we could get the probability of the location $i$ using Equation (2).

$$p_i = 1/(1 + e^{\beta_0 + \beta_1 * X_1 + \beta_2 * X_2 + ... + \beta_n * X_n}) \tag{2}$$

### 2.4. Gradient Boosting Decision Tree

In order to solve the problem that the traditional decision tree is prone to over-fitting, scholars combine the decision tree algorithm with the ensemble learning algorithm (bagging and boosting). The GBDT algorithm is one of the representative algorithms of boosting, which is also known as MART (multiple additive regression tree) or GBRT (gradient boosting regression tree). GBDT is one of the best algorithms for fitting real distributions in traditional machine learning algorithms, which has a strong generalization ability and can be used for classification problems or regression problems. It can also use the regularization function to correct the training results and reduce the degree of over-fitting.

The gradient boosting algorithm was proposed by Stanford statistics Professor Fridman in 2001, which is an approximation method using the gradient descent method [45].

The GBDT method is a member of boosting family in ensemble learning. Unlike the AdaBoost method, GBDT uses the CART regression tree as a weak classifier.

### 2.5. Imbalanced Sample Problem and Weighted GBDT Method

An imbalanced sample problem means the number of samples in the data set under each category varies greatly (difference in magnitude). For example, in the study of this paper, the number of non-landslide samples was 19-times that of landslide sample data. The usual practice to deal with the imbalanced data is randomly sampling non-landslide samples so that the landslide samples have the same order of magnitude as the non-landslide samples. Therefore, the non-landslide sample data are often wasted, resulting in poor prediction ability of the model. For the problem of sample imbalance, it can be solved from two aspects: data and algorithm.

At the data level, it is possible to oversample positive samples (landslide samples) or undersample negative samples (non-landslide samples). The current popular oversampling method is the Synthetic Minority Over-sampling Technique (SMOTE) method [46]. Borderline-SMOTE is an improved algorithm of SMOTE, which could solve the problem of sample overlap in the SMOTE algorithm [47]. The popular undersampling technique is the EasyEnsembletechnique [48]. The EasyEnsemble method divides the negative samples into multiple subclasses; all of the subclasses are trained, and then, the training scores are comprehensively analyzed to obtain the final classification result.

At the algorithm level, the cost matrix is used to set the weights corresponding to different categories, so that the misclassification cost of the negative samples is greater than that of the positive samples. The methods used in this article could fall into this category.

*2.6. Model Evaluation*

In this study, we used two statistical indices, namely sensitivity and specificity, to evaluate the performance of LSM models. Sensitivity and specificity are the proportion of landslide pixels that are correctly classified as landslide occurrences and the proportion of the non-landslide pixels that are correctly classified as non-landslides, respectively [35], which can be calculated with the following equations:

$$Sensitivity = \frac{TP}{TP + FN} \tag{3}$$

$$Specificity = \frac{TN}{FP + TN} \tag{4}$$

where TP (true positive) and TN (true negative) are the numbers of pixels that are correctly classified, while FP (false positive) and FN (false negative) are the numbers of pixels erroneously classified.

Based on the sense of sensitivity and specificity, we used the receiver operator characteristics (ROC) curve, the area under the ROC (AUC), and the recall value as evaluation indicators to evaluate the ability of the LSM models. The ROC curve is always used to evaluate the performance of diagnostic signals such as land changes [49,50], which uses sensitivity as the Y-axis against specificity as the X-axis with various cut-off thresholds [51]. The AUC represents the capability of a model to predict landslide and non-landslide cells. An AUC value of 1 indicates a perfect model, and 0 for a non-informative model [52]. In addition, we discussed the applicability of the accuracy value under the imbalanced conditions of landslide samples. The calculation equation for recall value and accuracy value is as follows:

$$Recall = \frac{TP}{TP + TN} \tag{5}$$

$$Accuracy = \frac{TP + TN}{TP + FN + TN + FP} \tag{6}$$

In order to find the best weight, we set the weight of the non-landslide sample to 1 and then increased the weight of the landslide sample from 1–30. To find an appropriate evaluation indicator, we introduced the balanced accuracy score, which means the average of the recall values for different categories of samples.

The choice of category weight can be seen in Figure 5. The best weight makes the balanced accuracy score the largest. We can also see that as the weight of the landslide sample increases, the value of recall increases gradually, while the value of AUC does not change much.

Table 2 shows the cost matrix used in this study.

**Table 2.** Cost matrix used in this study.

| True Label \ Predicted Label | Non-Landslide | Landslide |
|---|---|---|
| Non-landslide | 0 | 1 |
| Landslide | 17 | 0 |

From the table, we can find that the penalty coefficient of the correct classification is 0, which means that if the classification result was correct, no penalty would be imposed. The penalty coefficient of the non-landslide sample divided into landslide samples is 1. The penalty coefficient of dividing the landslide into non-landslide samples is 17. That means the cost of the misclassification of non-landslide samples is 17-times the cost of the misclassification of landslide samples.

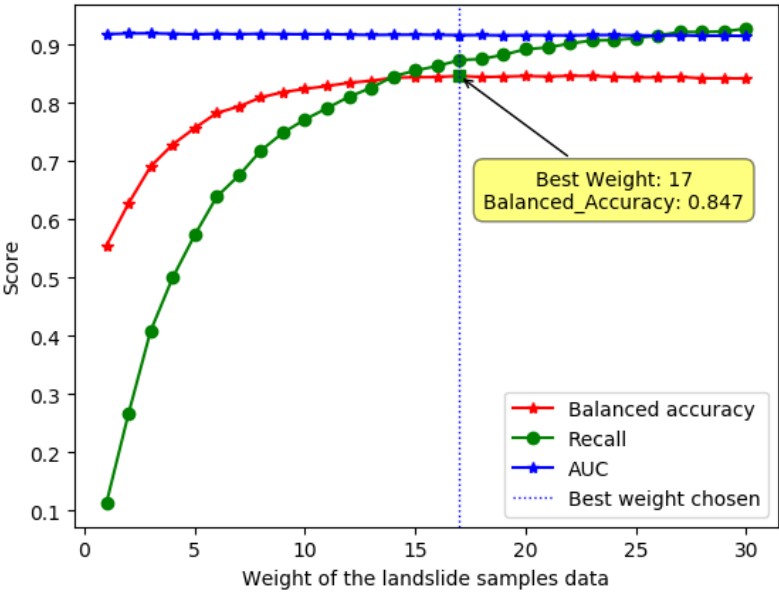

**Figure 5.** Balanced accuracy score, recall score, and AUC score with different weights.

## 3. Results

### 3.1. The Importance of Landslide Factors

In this study, we used the sklearn library [40] to calculate the importance of the landslide factors. The feature importance can be seen in Figure 6.

From Figure 6, we can find that the rainfall had the greatest impact on the landslide occurrence in the study area. The Seismic intensity and the distance from the river had almost the same impact. The impact of precipitation and river distribution on the landslide in the study area accounted for a large proportion.

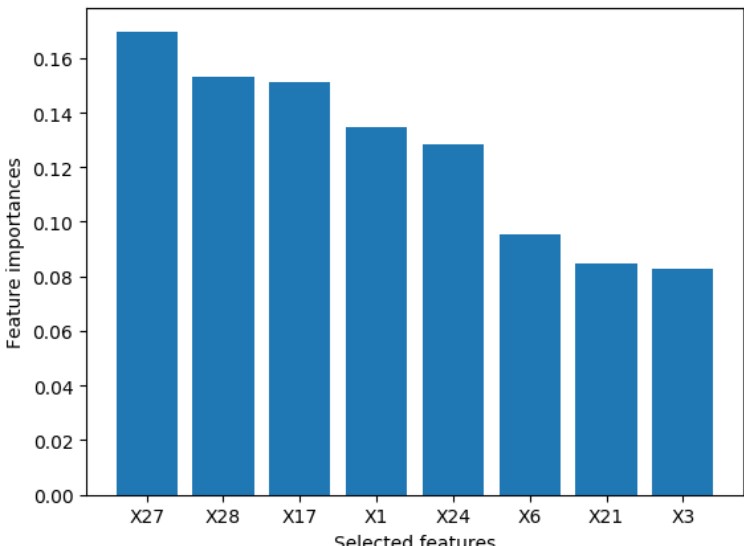

**Figure 6.** Feature importance of the landslide factors. X27: rainfall; X28: seismic intensity; X17: distance from the river; X1: elevation; X24: vertical distance from the river; X6: terrain surface texture; X21: depth of valley; X3: slope aspect.

### 3.2. Landslide Susceptibility Mapping Results

Figure 7 shows the results of using the three models for landslide susceptibility mapping, which were classified from the landslide prediction index (LPI). The higher the LPI is, the more likely the landslide will occur. Imbalanced classes are also the reason for not using the 0.50 decision threshold to separate classes for the three models. For an imbalanced learning problem, the value of the cut-off should consider the probability distribution of the classes (http://www.svds.com/learning-imbalanced-classes/). In this study, the natural breakpoint method was used to classify the LPI into five categories, which considers the histogram of the probability distribution for class division. The susceptibility map results by the three models are shown in Figure 7a–c.

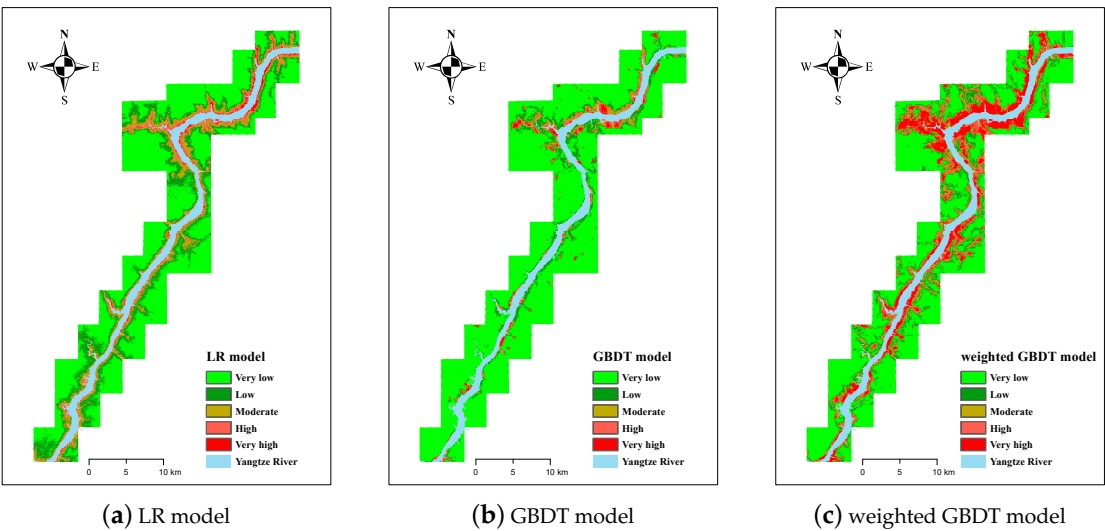

(**a**) LR model　　　　　　　　(**b**) GBDT model　　　　　　　(**c**) weighted GBDT model

**Figure 7.** LSM results of three models. (**a**) LSM using the LR model. (**b**) LSM using the GBDT model. (**c**) LSM using the weighted GBDT model.

As can be seen from Figure 7, the weighted GBDT model tended to divide more regions into very high levels, while the LR model tended to divide more regions into low and very low levels, and the GBDT model was somewhere in between. For a single image, the closer to the Yangtze River, the higher the landslide susceptibility.

Figure 8 shows the proportion of each category in each model.

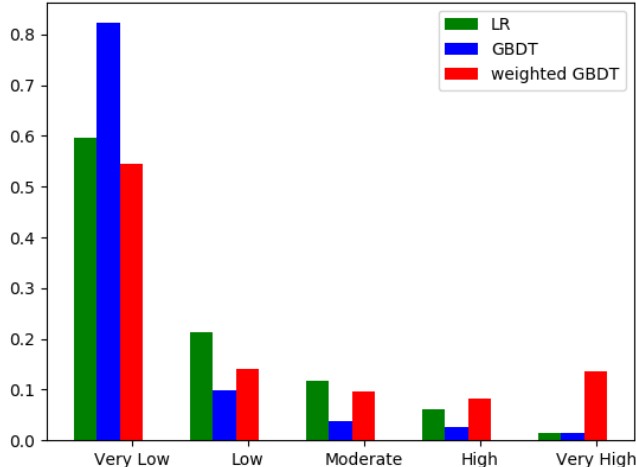

**Figure 8.** Distribution ratio of different landslide susceptibility classes for the LR, GBDT, and weighted GBDT models.

The reason for much more areas predicted as susceptible areas by weighted GBDT is as follows: For the weighted GBDT method, the weight of the landslide sample was greatly improved, so that the proportion of the landslide sample was also improved during the training of the model. Thereby the model's ability to predict landslide samples was enhanced.

### 3.3. Verification and Comparison

For the LR model, the LPI ranged from 0–0.783. The landslide susceptibility map was reclassified into five classes, that is very low (0–0.034), low (0.034–0.09), moderate (0.09–0.167), high (0.167–0.281), and very high (0.281–0.783), using the natural break method.

For the case of the GBDT model, the LPI ranged from 0–1. The landslide susceptibility map was reclassified into five classes, that is very low (0–0.055), low (0.055–0.182), moderate (0.182–0.370), high (0.370–0.609), and very high (0.609–1), using the natural break method.

For the weighted GBDT model, the LPI ranged from 0–0.991. The landslide susceptibility map was reclassified into five classes, that is very low (0–0.113), low (0.113–0.315), moderate (0.315–0.560), high (0.560–0.808), and very high (0.808–0.991), using the natural break method.

The ROC curve and the recall value were chosen to evaluate the predictive capabilities of the models. The reason for using the recall value is that the ROC curve could not fully describe the details of the predictive ability of the model [53]. The confusion matrix was used to compare the model prediction accuracy (Table 3). From Table 3, we found that that the accuracy scores of three models were very close, the accuracy of the three models reached 0.949 (LR), 0.968 (GBDT), and 0.953 (weighted GBDT), respectively. On the surface, it seems that all three models are very good. However, observing the percentage column, we can find that the LR model predicts only 23% of the landslide samples. This phenomenon is caused by the imbalance of landslide sample data. In this paper, the ratio of the landslide sample to the non-landslide sample reached 1:19, so that even if all of the landslide sample was misclassified, the misclassification rate of the landslide sample was only 0.05, which will not have a great impact on the accuracy value. Therefore, the accuracy value was not suitable as an evaluation indicator for sample imbalance. In this study, the ROC curve, the area under the ROC curve (AUC), and the recall value were used as the evaluation indicators, which are shown in Figure 9 and Table 4. The LR model had the lowest performance in terms of recall value and AUC, with values of 0.004 and 0.845. The GBDT model and weighted GBDT model exhibited a higher recall value and AUC value than the LR model, with recall values of 0.426 and 0.823 for the GBDT and weighted GBDT models and AUC values of 0.976 and 0.977 for GBDT and weighted GBDT models, respectively. The weighted GBDT model exhibited the highest AUC value and recall value among the three models. Therefore, the weighted GBDT method had the best predictive ability in landslide susceptibility mapping.

**Table 3.** The confusion matrix of the LR, GBDT, and weighted GBDT.

| Method | True Label | Predicted Label | | Percentage | Accuracy |
|---|---|---|---|---|---|
| | | **No** | **Yes** | | |
| LR | No | 165,841 | 111 | 95% | 0.949 |
| | Yes | 8760 | 34 | 23% | |
| GBDT | No | 165,382 | 570 | 97% | 0.968 |
| | Yes | 5045 | 3749 | 87% | |
| weighted GBDT | No | 133,049 | 32,903 | 95% | 0.953 |
| | Yes | 34 | 8760 | 76% | |

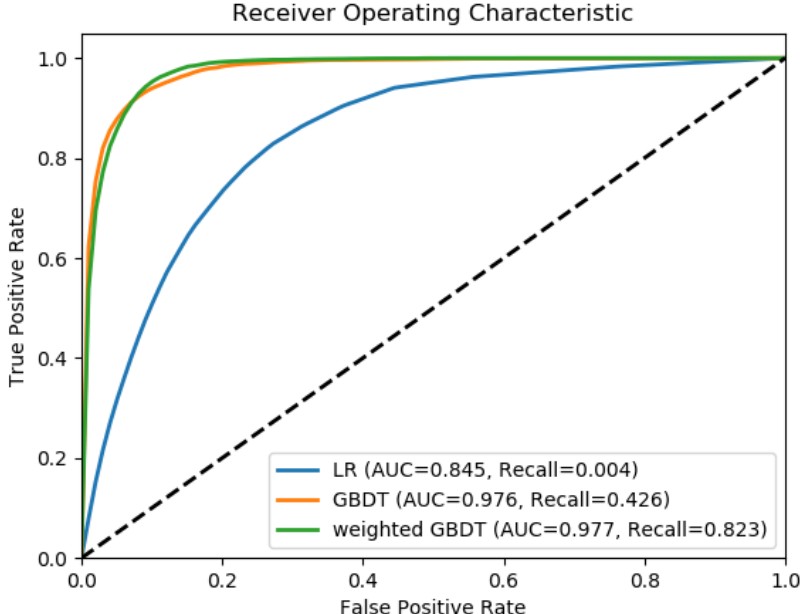

**Figure 9.** The ROC curve of the three LSM models.

**Table 4.** AUC and recall value for the three LSM models.

| Parameters | LR | GBDT | Weighted GBDT |
|------------|-------|-------|---------------|
| AUC | 0.845 | 0.976 | 0.977 |
| Recall | 0.004 | 0.426 | 0.823 |

An actual landslide location verification of three models can be seen in Figure 10. It can be seen from Figure 10a that the LR model is not very good at distinguishing the real landslides. Only the landslides near the Yangtze River, such as Huangnibao landslide, Kangjiapo landslide, Baiyutuo landslide, and Shiguiliang landslide, can be identified as very high-susceptibility areas and high-susceptibility areas. The Datuancun landslide and the Guankou landslide are identified as the low-susceptibility area or very low-susceptibility area, which is not consistent with the actual situation.

Figure 10b shows a better result than the LR model. Most of the landslide areas were classified into high-susceptibility and very high-susceptibility areas. However, most areas of the Guankou landslide were classified into low-susceptibility areas. The Taojiaping landslide, No. 2 Solid waste landslide, and No. 3 Solid waste landslide (mixture of historical landslide material and solid waste) were classified into moderate-susceptibility area.

Figure 10c shows the best classification result. Almost all of the landslides were classified into high-susceptibility areas and very high-susceptibility areas, which also proves the effectiveness of the weighted GBDT method.

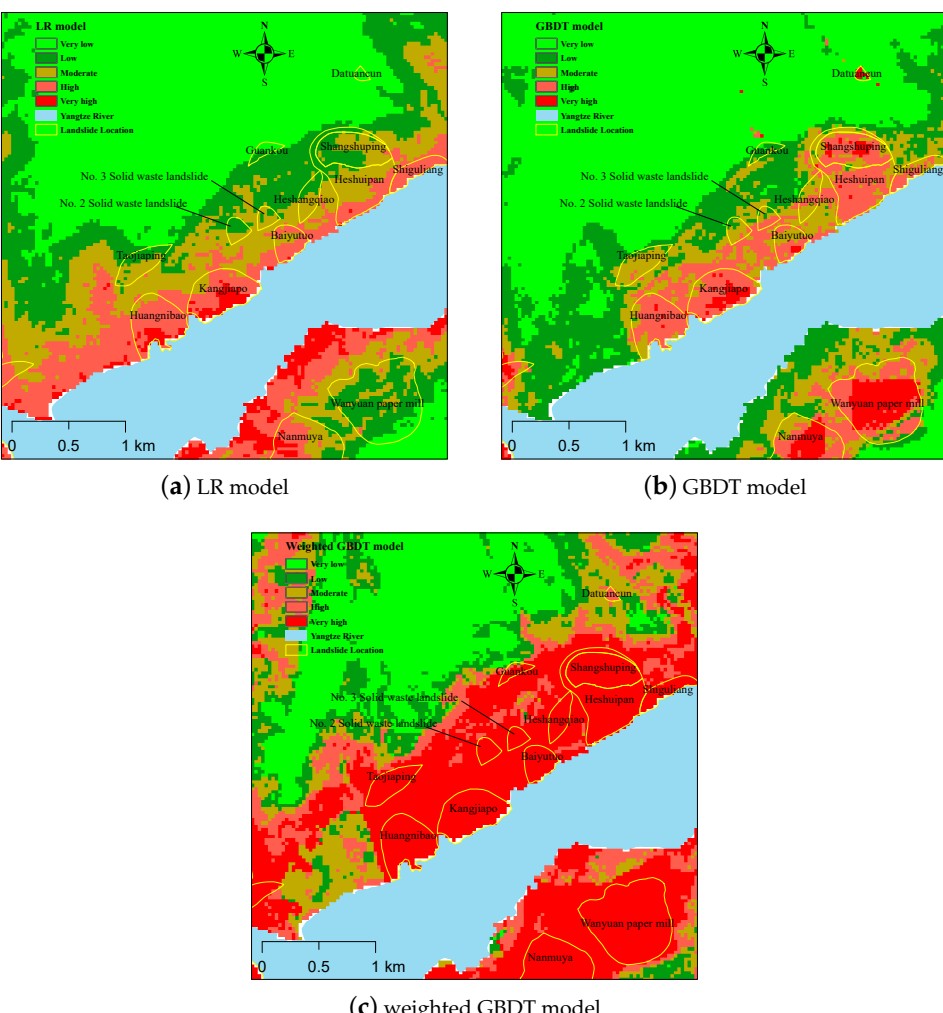

(**a**) LR model                    (**b**) GBDT model

(**c**) weighted GBDT model

**Figure 10.** Actual landslide location verification of the three models. (**a**) Actual landslide location verification using the LR model. (**b**) Actual landslide location verification using the GBDT model. (**c**) Actual landslide location verification using the weighted GBDT model.

## 4. Discussion

Landslide susceptibility could be analyzed as a regression problem or a classification problem. When treated as a classification problem, more and more machine learning methods are used as classifiers in the current studies. However, in these studies, the landslide susceptibility evaluation problem was always described as a normal binary classification problem rather than an imbalanced learning problem. Based on the previous studies, this paper considered the problem of imbalanced landslide samples and describes the LSM problem as an imbalanced learning problem.

In this study, we analyzed the problems caused by the imbalance of landslide samples and found that the accuracy value cannot evaluate and distinguish the pros and cons of the model very well because the accuracy value indicates the overall accuracy of the model, including the landslide samples and the non-landslide samples. When the number of non-landslide samples is much larger than that of the landslide samples, the cost of the landslide sample misclassification becomes insignificant. According to the calculation equation of the accuracy (Equation (6)), assuming that the number of non-landslides is infinite, the value of the accuracy value is infinitely close to one, rather than zero, which proves that the accuracy value is not suitable for landslide susceptibility evaluation when the sample is not balanced. Imbalanced sample data may lead to a biased prediction of the negative class, that is non-landslide class.

Therefore, we introduced the recall value into the evaluation of the models, because the recall value only pays attention to how many positive samples are correctly distinguished. At the same time, in the landslide susceptibility evaluation, the focus of our attention is exactly how many landslide samples are correctly predicted, which coincides with the meaning of the recall value representatives. It can be seen from the recall value calculation equation (Equation (5)) that the more the landslide sample is predicted, the higher the recall value will be.

The LR model shows a relatively high accuracy value (0.949), a moderate AUC value (0.845), and a very low recall value (0.004), which proves that the landslide sample has a very large misclassification ratio, while the non-landslide sample has a very small misclassification ratio. According to the previous analysis, we can assert that the LR model is not a good model under the condition that the landslide sample is not balanced. Even the accuracy value and AUC value of the model are very high.

The accuracy value (0.968), AUC value (0.976), and recall value (0.426) of the GBDT model are higher than those of the LR model, indicating that the performance of the GBDT model is better than the LR model. This is mainly because the ensemble learning method uses multiple weak classifiers to analyze and evaluate the classification results.

The weighted GBDT model has a very high accuracy value (0.953), the highest AUC value (0.977), and recall value (0.823) in the three models, which shows that by setting different weights for different categories, the prediction performance of the model can be improved. The significant increase in recall value also demonstrates the good performance of the weighted GBDT method in dealing with landslide sample imbalance problems, meaning the landslide samples could be distinguished correctly much more than the other two models.

The key to the weighted GBDT method is the choice of sample weights. However, the selection of the best sample weights requires iteratively calculating the value of the balanced accuracy score under different weights, which would take a lot of time. Therefore, the rapid search for the best weight remains to be studied.

It is worth noting that we only discussed the solving of the imbalance of landslide samples problem at the algorithm level. The solution at the data level is still worth exploring and researching; for example, using the EasyEnsemble method to downsample non-landslide data, using the SMOTE method to oversample landslide samples, etc. In addition, more evaluation indicators for imbalanced learning need to be taken into account, such as precision, F1-score, etc.

## 5. Conclusions

In this study, by applying the LR model, the GBDT model, and the weighted GBDT model to the landslide susceptibility mapping, we can get the following conclusions:

A novel model of LSM was proposed named weighted GBDT that is more suitable for LSM than LR model and the GBDT model when the landslide sample is not balanced.

Compared with previous studies, this paper focused on the problems caused by the imbalance of landslide samples. The problem of landslide susceptibility evaluation is no longer regarded as an ordinary binary classification problem, but an imbalanced learning problem. At the algorithm level, the imbalanced learning problem was processed and analyzed using the weighted GBDT method. Finally, for this imbalanced learning problem, the recall value was used as the evaluation index instead of the accuracy value.

In most cases in practice, the number of landslide samples is often much smaller than non-landslide samples, so the landslide sample imbalance is a common problem. The method proposed in this paper is expected to play a greater role in the process of landslide disaster risk management.

**Author Contributions:** Conceptualization, Y.S. and R.N.; methodology, S.X.; software, T.G.; validation, L.P.; investigation, R.Y.; resources, S.X.; writing, original draft preparation, Y.S.; writing, review and editing, S.X.; project administration, S.L.; funding acquisition, R.Y., L.P. and T.C.

**Funding:** This research was jointly funded by the Geological Survey Project (No. 0431203), the Three Gorges Follow-up Work on Geological Disaster Prevention and Research Project (No. 0001212018CC60010, 0001122012AC50021), the National Natural Science Foundation of China (No. 41602362) and the National Natural Science Foundation of China (No. 61601418).

**Acknowledgments:** Thanks to Wang Zhensheng and Wang Jiqing for the help provided during the article writing process. We used Python and the following packages: matplotlib, numpy, pandas, hyperopt, scipy, sklearn, and pickle. Some of the images in the text were drawn using the matplotlib library; the others were drown with ArcGIS 10.2 and Office 2016. The paper was written using the Latex plugin in Visual Studio Code. Furthermore, we would like to express our gratitude to USGS for the Landsat-8 image data and to NASA for the ASTER GDEM data.

**Conflicts of Interest:** The authors declare no conflict of interest.

## Abbreviations

The following abbreviations are used in this manuscript:

| | |
|---|---|
| AUC | area under the ROC curve |
| CART | classification and regression tree |
| CF | certainty factor |
| DEM | digital elevation model |
| GBDT | gradient boosting decision tree |
| GIS | geographic information system |
| LR | logistic regression |
| LSM | landslide susceptibility mapping |
| LPI | landslide prediction index |
| NDVI | normalized difference vegetation index |
| NDWI | normalized difference water index |
| PC | Pearson correlation |
| PCA | principal components analysis |
| RF | random forest |
| RFE | recursive feature elimination |
| ROC | receiver operating characteristics |
| RS | remote sensing |
| RSI | river strength index |
| RSP | relative slope position |
| SMOTE | synthetic minority over-sampling technique |
| TCI | terrain convergence index |
| THI | terrain humidity index |
| TPI | terrain position index |
| TRI | terrain roughness index |
| TSC | terrain surface curvature |
| TST | terrain surface texture |
| USGS | United States Geological Survey |
| VDR | vertical distance from the river |

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
