# Peer review of "Landslide Susceptibility Mapping Based on Weighted Gradient Boosting Decision Tree in Wanzhou Section of the Three Gorges Reservoir Area (China)"

_ijgi, doi:10.3390/ijgi8010004_

Round 1

Reviewer 1 Report

The authors studied Landslide susceptibility mapping based on weighted gradient boosting decision tree in Wanzhou section of the Three Gorges reservoir area (China). They integrated weighted gradient boosting decision tree models and a logistic regression model to validate the landslide susceptibility. This paper highlights the difference between the three types of analysis approach and gives the reader essential information about the scientific value. The manuscript is abundant in the contents of materials for the proposed model, and just a few unclear descriptions remain in the draft. I think it can be accepted for publication in ISPRS Int. J. Geo-Information after minor revision.

These points are as follows:

1. Abstract and introduction: The draft needs to add some new publications related to regional landslide susceptibility assessment in this area. Also, please introduce very carefully the innovative of this research compared with previous publications in the abstract.

2. Fig.1 and Fig.2: please add the hill-shading map on both pictures. In this way, the reader may understand the regional topography more preciously.

3. Discussion: The proposed analysis approach review in the draft is plentiful. However, the limitation of these methods should be mentioned explicitly and 

simply.

4. I suggest the English must be polished by a native English speaker because there are some grammatical mistakes and somewhat strange expressions in the draft.

Author Response

Thanks very much for you advice. My answer is marked as red in the word file No.1.docx

Reviewer 2 Report

General comments to authors 
There are thousands of Landslide susceptibility mapping papers. In general, the topic of the manuscript is a bit interesting. However, the introduction is a bit surficial, please deepen you’re the research.  Some extra information should be added and careful corrections should be made (suggestions are presented below).

1.         ROC first appear in the abstract, Better to explain it and help reader easily understand.

2.         The authors should clarify if they used a normalization procedure or simple used the factor according to their classification. If they used them according to their classification please comment if this has an influence on their performance. 

3.         There are many equations in method, because these equations were proposed by the others, better to make it concise.

4.         The authors should clarify if they used a normalization procedure or simple used the factor according to their classification. If they used them according to their classification please comment if this has an influence on their performance. 

5.         I also suggest the authors to include in their study, a multi-collinearilty or similar analysis to illustrate the predictive capability of each factor and the necessity to use them, or else try to provide an explanation why this procedure may not be necessary. Sometimes, not more factors, mean good accuracy, please refers to the followed references (An integrated artificial neural network model for the landslide susceptibility assessment of Osado Island, Japan) because the authors used 29 factors.

6.          I also suggest performing an analysis on the existence of statistical significant variation or not between the performances of the used models by conducting appropriate statistical tests.

7.          In my opinion, a further comparison between the three methods should be established and a more comments about previous studies.

8.         For the fig3, the authors should describe the similarly and difference LSM produced by LR, GBDT, and weighted GBDT, not only the maps.

9.         As for Fig.4, why there are much more areas predicted as susceptible areas by weighted GBDT? Maybe the readers want to know some details.

10.     Please clarify the type of landslide, because the triggering mechanism is different.

11.     Table 2 and table3 are very similar, it is a bit repeated, please combine together. 

12.     The paper is full of jargon, has too many different forms of analysis and doesn't have a presentation format that is typical or intuitive for the presentation of results. The authors need to drastically reduce complexity of the results section if a general audience is going to understand it. 

13.     In the results section that is done by focusing the writing of the results on the biological responses with references to statistics in parentheses or tables.  There is too much statistical jargon in these results as now written.

Some grammar mistake, such L24, the verb of cause should be different way, please confirm it and check it carefully.

Author Response

Thanks very much for your advice. My answer is marked as red in the word file No.2.docx.

Reviewer 3 Report

Dear authors, I have read and carefully evaluated your mauscript. I think it could be published after some moderate revisions. Please, find below my comments.

GENERAL COMMENTS

English is quite good but I recommend a thorough check by an expert English speaker, to polish the language and to remove small errors.

The test site should be described more thoroughly. E.g. geomorphological features and processes should be better constrained and the landslides should be better characterized (e.g. I did not find a clear description of their typology and their mechanism). Moreover, avoid reporting the meteorological statistics (e.g. temperature and rainfall) of a given year: they may not be representative. Instead, you should show the mean values averaged over long time series (mean temperature, mean annual precipitation and so on).

There is one thing that is not clear in the methodology. You used many variables (namely, 29). How did you addressed problems of multicollinearity and mutual interactions? Moreover, are you sure all of them are useful? Usually when so many parameters are used, a forward selection of the most influential parameter is performed, and some of them are found to be pejorative and are thus discarded. Have you considered this issue?

The idea of addressing the problem of imbalance between landslide and non-landslide data is interesting. However, I have a few comments and I would like you to address them in the revised manuscript. First, the choice of the weights 5 and 1 (cost matrix in Table 2) seems arbitrary and subjective. What happens if other weights are assigned? Second, I have concerns about the definition of the recall function. From equation 14 it seems exactly the same than Sensitivity (eq. 12). Third, what happens to the skill scores if the imbalance is not considered? Fourth, can you provide a comparison of the skill scores (Auc, recall) values when calculated according to the weights of table 2, no weights and some othe rweights? This would be very interesting and would allow a more complete discussion.

SPECIFIC COMMENTS

Line 25. References 1 and 2 do not seem very pertinent. Please, cite some more general work on landslide hazard/damages at global scale. E.g. You can cite Petley (2012).

Line 25-28. This paragraph should be moved to the test site description (especially the part describing water level fluctuations). Please change “have” into “has”.

Line 30. Please, consider also citing Trigila et al. (2013)

Line 37. Please, consider also citing Manzo et al. (2013)

Line 51. Here you can add a few lines about your study area. You can move here the lines 25-28 (except the water level fluctuation that is more consistent in the test site description).

Line 57. Please, write the areal extension of your study area.

Lines 61-62. Please, rephrase. What does it mean “vertical” and “horizontal” referred to a river? Maybe that the direction is N-S and W-E?

Line 64-65. Please, consider rephrasing. Maybe the sentence flows better if you say that from a geological point of view the area is mainly composed by Triassic and Jurassic formations.

Line 77. What scale is the geological map used?

Line 101. What does it mean “optimize the parameters”? It is very generic, please be more precise on what you did and why.

Table 1. I suggest moving “data description” column immediately after the “name” column. In this way, it provides a quick reference in case the reader is not familiar with one or more abbreviations.

Line 211. I don’t agree on the assumption that a landslide occurs when LPI>0.5. The meaning of the landslide susceptibility percentages is more difficult to be interpreted, and thus the choice of the break values is very important and difficult and it changes the meaning of each class. I find that the natural break method may e weak, because it does not account for the distribution of susceptibility values inside and outside the landslide areas, 

Line 216. I suggest to add in this section the figures A1, A2 and A3, merged into a single figure. They are very pertinent and I suggest to add them to the main text.

Line 144. What do you mean with “solid waste landslide”? Is it made by generic waste material (garbage) or by remains of mining activity?

Line 252. Actually, in some cases landslide susceptibility is analyzed as a regression problem (not to classify stable and unstable areas, but e.g. to predict the landslide density).

Line 255. Your analysis is not so accurate. It would be much more complete if you explore other possibilities and the impact of the matrix cost on the recall and AUC values, how I suggested in my last general comment.

Line 292-293. Please, move this sentence to the end of the manuscript.

CITED REFERENCES

Manzo, G., Tofani, V., Segoni, S., Battistini, A., & Catani, F. (2013). GIS techniques for regional-scale landslide susceptibility assessment: the Sicily (Italy) case study. International Journal of Geographical Information Science, 27(7), 1433-1452.

Petley, D. (2012). Global patterns of loss of life from landslides. Geology, 40(10), 927-930.

Trigila, A., Frattini, P., Casagli, N., Catani, F., Crosta, G., Esposito, C., ... & Spizzichino, D. (2013). Landslide susceptibility mapping at national scale: the Italian case study. In Landslide science and practice (pp. 287-295). Springer, Berlin, Heidelberg.

Author Response

Thanks very much for your advice. My answer is marked as red in the word file No.3.docx.

Round 2

Reviewer 2 Report

This version improved, however, some parts still need to be revised before publication. Such as Q5, as I said before, the authors used 29 factors, it is better to do multi-collinearilty, As Guzzetti (1999), not more factors, it got good results. For the multi-collinearilty analysis, you may use forward selection, you can refer the following references (Dou et al., 2015) for Collinearity Statistics in this paper.

Guzzetti, F., Carrara, A., Cardinali, M., Reichenbach, P., 1999. Landslide hazard evaluation: A review of current techniques and their application in a multi-scale study, Central Italy, in: Geomorphology. pp. 181–216. doi:10.1016/S0169-555X(99)00078-1

Optimization of Causative Factors for Landslide Susceptibility Evaluation Using Remote Sensing and GIS Data in Parts of Niigata, Japan. PLOS ONE 10, e0133262. doi:10.1371/journal.pone.0133262

Q6 still not reply, maybe do statistical significant whether these used model whether there are significant among each other.

Q10 please clarify show the main types of landslides, collapsed deposit is a bit strange.

Also, it is better to add a fig including 29 factors for readers to understand the factors in the local area. 

Author Response

Thank so much for you advice. My answer is in No2-new.docx

Reviewer 3 Report

Dear Authors, I think the manuscript has greatly improved and it is now ready for publication.

Just a few remarks:

- Please, check the manuscript and be sure that you define all acronyms (e.g.: PCA at line 91)

- Maybe I'm mistaken, but I think you didn't clearly describe what kind of landslides affect you study area. It is important to know the landslide typology for a susceptibility study.

- About reducing the variable space from 29 to 8 parameters: which are they? can you highlight them somehow in table 1 or in the text? Are they the same for each of the three models?

- About LPI classes definition, I found that the modified text is better than the previous version. About defining objective and robust classification methods, I personally use other methods (see e.g. Segoni et al., 2015, Fig. 6). I do NOT mean that you have to use this method as well, it is just to exchange ideas and approaches. 

Segoni, S., Lagomarsino, D., Fanti, R., Moretti, S., & Casagli, N. (2015). Integration of rainfall thresholds and susceptibility maps in the Emilia Romagna (Italy) regional-scale landslide warning system. Landslides, 12(4), 773-785.

Author Response

Thanks for your advice so much. My answer is in No3-new.docx
